# Impact of COVID-19 Vaccination on Healthcare Worker Infection Rate and Outcome during SARS-CoV-2 Omicron Variant Outbreak in Hong Kong

**DOI:** 10.3390/vaccines10081322

**Published:** 2022-08-15

**Authors:** Sze-Tsing Zee, Lam-Fung Kwok, Ka-Man Kee, Ling-Hiu Fung, Wing-Pan Luk, Tsun-Leung Chan, Chin-Pang Leung, Pik-Wa Yu, Jhan Hung, Kit-Ying SzeTo, Wai-Leng Chan, Siu-Fai Tang, Wai-Chi Lin, Shiu-Kwan Ma, Koon-Hung Lee, Chor-Chiu Lau, Wai-Hung Yung

**Affiliations:** 1Department of Pathology, Hong Kong Sanatorium & Hospital, Hong Kong Special Administrative Region, Hong Kong 999077, China; 2Infection Control Team, Hong Kong Sanatorium & Hospital, Hong Kong Special Administrative Region, Hong Kong 999077, China; 3Quality & Safety Division, Hong Kong Sanatorium & Hospital, Hong Kong Special Administrative Region, Hong Kong 999077, China; 4Research Department, Hong Kong Sanatorium & Hospital, Hong Kong Special Administrative Region, Hong Kong 999077, China; 5Hospital Administration, Hong Kong Sanatorium & Hospital, Hong Kong Special Administrative Region, Hong Kong 999077, China

**Keywords:** SARS-CoV-2, Omicron variant of concern, homologous boosting, heterologous boosting, CoronaVac, BNT162b2, healthcare worker, return-to-work

## Abstract

Immune escape is observed with SARS-CoV-2 Omicron (Pango lineage B.1.1.529), the predominant circulating strain worldwide. A booster dose was shown to restore immunity against Omicron infection; however, real-world data comparing mRNA (BNT162b2; Comirnaty) and inactivated vaccines’ (CoronaVac; Sinovac) homologous and heterologous boosting are lacking. A retrospective study was performed to compare the rate and outcome of COVID-19 in healthcare workers (HCWs) with various vaccination regimes during a territory-wide Omicron BA.2.2 outbreak in Hong Kong. During the study period from 1 February to 31 March 2022, 3167 HCWs were recruited, and 871 HCWs reported 746 and 183 episodes of significant household and non-household close contact. A total of 737 HCWs acquired COVID-19, all cases of which were all clinically mild. Time-dependent Cox regression showed that, compared with two-dose vaccination, three-dose vaccination reduced infection risk by 31.7% and 89.3% in household contact and non-household close contact, respectively. Using two-dose BNT162b2 as reference, two-dose CoronaVac recipient had significantly higher risk of being infected (HR 1.69 *p* < 0.0001). Three-dose BNT162b2 (HR 0.4778 *p*< 0.0001) and two-dose CoronaVac + BNT162b2 booster (HR 0.4862 *p* = 0.0157) were associated with a lower risk of infection. Three-dose CoronaVac and two-dose BNT162b2 + CoronaVac booster were not significantly different from two-dose BNT162b2. The mean time to achieve negative RT-PCR or E gene cycle threshold 31 or above was not affected by age, number of vaccine doses taken, vaccine type, and timing of the last dose. In summary, we have demonstrated a lower risk of breakthrough SARS-CoV-2 infection in HCWs given BNT162b2 as a booster after two doses of BNT162b2 or CoronaVac.

## 1. Introduction

As of the end of June 2022, SARS-CoV-2 has caused over 500 million cumulative cases of COVID-19 and over 6 million deaths according to the World Health Organization, but the actual global death toll could be millions more than the official counts [1,2]. Vaccination is considered the most important tool in controlling the pandemic. The Omicron (Pango lineage B.1.1.529) variant of concern (VOC) emerged in November 2021 in South Africa and soon became the predominant circulating strain worldwide, replacing Alpha, Beta, Gamma, and Delta, which are now categorized as “previously circulating VOC” [1]. Waning immunity after vaccination and immune escape from Omicron VOC have rendered various vaccine platforms less effective [3,4,5]; however, homologous and heterologous boosting was shown to restore immunity against infection by raising neutralizing activity and T-cell response [6,7,8,9].

Two types of vaccines have been available in Hong Kong since late February 2021: inactivated COVID-19 Vaccine (CoronaVac; Sinovac) and mRNA Vaccine (BNT162b2; Comirnaty). As a result of vaccine hesitancy in the general public, Hong Kong has been severely hit by Omicron, predominantly BA.2.2, since late January 2022—“the fifth wave”. Daily new cases surged exponentially from few hundred in early Feb to over 70,000 in early March, overwhelming both routine and emergency medical care as well as isolation facilities [10,11,12,13]. By the end of January 2022, 1.2 million infections were reported in Hong Kong during the fifth wave, which resulted in over 9000 deaths, with the majority being elderly with incomplete or no vaccination [13,14].

HKSH Medical Group, with more than 3100 clinical and non-clinical healthcare workers, provides service to the public via a network of 600-bed acute hospitals (Hong Kong Sanatorium and Hospital, Hong Kong, China), 2 oncology centers, and 4 outpatient centers located on different parts of the Hong Kong Island. In response to the fifth wave, HKSH implemented a series of enhanced measures to prevent nosocomial SARS-CoV-2 transmission through (1) optimization of the staff vaccination rate, (2) enhancing COVID-19 surveillance (mandatory reverse transcription-polymerase chain reaction (RT-PCR) pre-admission screening for all patients and those who required mask-off procedures, and mandatory regular screening for staff using rapid antigen test (RAT), (3) stringent contact tracing and testing policies. We performed a retrospective study to evaluate the effect of COVID-19 vaccination on staff infection rate, their outcome, and their time to return to work. The study was approved by the Research Ethics Committee of the HKSH Medical Group (REC-2022-05).

## 2. Materials and Methods

### 2.1. Recruitment and Definitions

All full-time staff of HKSH Medical Group with no history of COVID-19 before 1 February 2022 were recruited. Their demographics, job category, and COVID-19 vaccination history were retrieved from employment records and hospital vaccine records. A case of COVID-19 was defined as a RT-PCR- or RAT-confirmed infection between 1 February and 31 March 2022.

To evaluate the effect of vaccination on the infection rate, the dose of vaccine given within the 14-day period before COVID-19 confirmation was disregarded. Incomplete vaccination was defined as the receipt of fewer than 2 doses, while the receipt of 2 or more doses was defined as fully vaccinated. Severe COVID-19 was defined as any case that required oxygen therapy or hospitalization.

For evaluation of the time to return to work, we only included staff who were fully vaccinated and diagnosed between 26 February and 31 March 2022. This is because prior to this period, all infected persons in Hong Kong were required to undergo 14-day isolation in community isolation facilities (CIF) or Hospital Authority (HA) hospitals as required by the Department of Health, HKSAR. From 26 February 2022 onwards, infected persons may discontinue isolation at their premises after 2 successive negative RATs on days 6 and 7, should they have received at least two doses of COVID-19 vaccines.

### 2.2. Data Collection and Follow-Up Testing for Confirmed Healthcare Worker

Confirmed COVID-19 cases were required to provide clinical information including symptoms, onset date, reasons for testing, RAT result (if performed), and exposure history via a standard online questionnaire. Upon resolution of fever and improvement in symptoms, fully vaccinated staff underwent RAT on 2 consecutive days, earliest on days 6 and 7 (day 0 = first specimen with positive RT-PCR). RT-PCR was performed on the 2nd day of negative RAT. For infected staff with incomplete vaccination, the earliest negative RAT results accepted for RT-PCR testing were on days 13 and 14. A negative RT-PCR or a positive RT-PCR with an E gene cycle threshold (Ct) value of 31 or above were used as criteria for return-to-work. If the cycle Ct value was less than 31, RT-PCR was repeated daily until it was 31 or above.

### 2.3. Staff Reporting Close Contact with Confirmed COVID-19 Cases

Staff who had exposure to confirmed COVID-19 were requested to inform the infection control team (ICT) for risk assessment. Those with significant exposure according to our infection control guideline (Appendix A) were offered serial RT-PCR on days 1, 4, and 8 (day 1 = exposure day) to rule out infection. Duty could be resumed if day 4 RT-PCR was negative but daily RAT was required till a negative RT-PCR was achieved on day 8.

### 2.4. Mandatory RAT COVID-19 Screening for Staff

RAT screening every 3 days (8–15 February 2022), every day (16–28 February 2022), on alternate days (1 April 2022 onward) was mandatory for all clinical and non-clinical staff before starting their duty. The RAT screening frequency was adjusted according to the intensity of transmission in the local community and a recommendation from our ICT. Staff with compatible symptoms but negative RAT were offered RT-PCR to rule out infection.

### 2.5. Rapid Antigen Test and Reverse Transcription-Polymerase Chain Reaction (RT-PCR)

RAT was performed exclusively using nasal swab by INDICAID^®^ COVID-19 Rapid Antigen Test, which is an immunochromatographic membrane assay intended for the qualitative detection of SARS-CoV-2 nucleocapsid antigens. The tests were performed according to the manufacturer’s recommendation and our previous publication [15].

SARS-CoV-2 RT-PCR was performed using combined nasal and throat swab by the detection of virus N gene, E gene, RdRp gene, S gene, M gene, or ORF1ab gene using different platforms including Abbott Alinity m, TIB MolBiol/FujiFilm Wako, coupled with Roche qPCR platforms, DiaSorin, Cepheid GeneXpert, and BioFire FilmArray. All SARS-CoV-2 positive specimens were confirmed by more than one platform and submitted to the reference laboratory for final confirmation. The tests were performed according to the manufacturers’ recommendation. Specimens from recovering HCWs were tested by Cepheid GeneXpert exclusively for E gene Ct value.

### 2.6. Statistical Analysis

Demographics, history of significant SARS-CoV-2 exposure, and rate of COVID-19 were tested using a t-test and Fisher’s exact test/Chi-squared test. Since vaccination was ongoing during the study period, the study was crossover in nature. To compare the effect of 3-dose, 2-dose group, and specific regimes, these variables were treated as a time-varying covariate. Time-dependent Cox regression model was used to model the dose effect on time to SARS-CoV-2 infection. Time-dependent Cox regression was computed using R software version 4.1.0 (R Foundation, Vienna, Austria) [16]. The hazard ratio plot was created by R package “survminer” [17].

## 3. Results

### 3.1. Demographics and Vaccination History

After excluding 8 staff members who had a history of COVID-19 before 1 February 2022 had been excluded, 3167 (2329, 73.6% female) were included in the analysis of vaccination effectiveness. By 1 February 2022, the first day of the study period, 2953 (93.2%) were regarded as fully vaccinated (received at least two doses). By 31 March 2022, the last day of the study period, 3103 (98.0%) had received at least two doses while booster doses (a third dose) were given to 1435 (45.3%). A total of 160 (5.1%) received heterologous boosting, while 983 (31.3%) and 291 (9.2%) received homologous boosting with BNT162b2 and CoronaVac, respectively (Table 1).

### 3.2. Breakthrough COVID-19 and Symptoms

During the study period (1 February–31 March 2022), 737 staff members acquired COVID-19, which accounted for 23.3% of all full-time employees. The majority were female (80.9%), with a mean age of 37.7 years. COVID-19 was confirmed by RAT alone in 298 (40.4%), RT-PCR alone in 220 (29.9%), and both RAT and RT-PCR in 219 (29.7%). New onset of COVID-19-related symptoms (53.8%) was the most common reason for testing that led to the diagnosis of COVID-19, followed by exposure history to a confirmed/suspected case (43.3%). At the time of data collection, the majority (*n* = 649, 88.1%) were symptomatic, with sore throat (81.1%), coughing (60.6%), and running nose (46.7%) being the most common symptoms. All of them had mild disease, and none required hospitalization (Table 2).

### 3.3. Significant SARS-CoV-2 Exposure, Vaccination Regime, and Risk of COVID-19

A total of 871 staff members (701, 80.48% female) reported 746 and 183 episodes of significant household and non-household close contact. There was no significant nosocomial exposure. Demographics, the nature of exposure, and the rate of COVID-19 stratified by vaccination regime are shown in Table 3. Three-dose regimes were associated with a lower incidence of COVID-19 than two-dose regimes. Ninety staff members who had incomplete vaccination (0–1 dose) were excluded from further analysis of vaccine effectiveness. Another five staff members with uncommon vaccine combinations were also excluded (note for Table 3).

Time-dependent Cox regression showed that three-dose vaccination reduced the risk of infection by around 50% (hazard ratio 0.5339 *p* < 0.0001) when compared with two-dose vaccination. Females had a significantly higher risk than males (HR 1.43 *p* = 0.0005), while age and job category (clinical vs. non-clinical) had no significant effects on infection risk. Close household contact was associated with the highest risk of infection (HR 4.81 *p* < 0.0001), while the risk from non-household close contact is only similar to those with no known close contact (Table 4 and Figure 1) Compared with two-dose vaccination, three-dose vaccination was found to reduce infection risk by 31.7%, 89.3%, and 58% in household contact, non-household close contact, and no known contact group, respectively (Table 5).

Further regression analysis (using two-dose BNT162b2 as reference) showed that two-dose CoronaVac had a significantly higher risk of being infected (HR 1.69 *p* < 0.0001). Three-dose BNT162b2 (HR 0.4778 *p* < 0.0001) and two-dose CoronaVac + BNT162b2 booster (HR 0.4862 *p* = 0.0157) were associated with lower risk of infection. Three-dose CoronaVac and two-dose BNT162b2 + CoronaVac booster were not significantly different from two-dose BNT162b2 (Table 6 and Figure 2)

### 3.4. Time to Achieve Negative RAT and RT-PCR Criteria for Return-to-Work

During the study period (26 February–31 March 2022), 422 recovering staff members, who were previously fully vaccinated, were included in the return-to-work analysis. The mean time taken to achieve two consecutive negative RAT was 9.76 days. Upon two consecutive negative RATs, only 310 (73%) fulfilled RT-PCR criteria (negative or E gene Ct value 31 or above) for return-to-work. (Figure 3) The mean time for return-to-work based on RT-PCR criteria was 10.1 days and was not affected by age, number of vaccine doses taken, vaccine type, and timing of the last dose (Table 7).

## 4. Discussion

To our knowledge, this is the first study comparing the efficacy of different combinations of mRNA and inactivated COVID-19 vaccines in healthcare workers. Our cohort is a relatively young population with a high vaccination rate. Since all of our staff acquired infection from the community, the incidence during the fifth wave mirrored the intensity of SARS-CoV-2 transmission in the general public. By the end of March 2022, the number of confirmed cases (by RT-PCR and RAT) in Hong Kong reached 1,164,138, which accounted for 15% of Hong Kong’s population [13]. Using mathematical modeling, local epidemiologists estimated that 4 million, or 60% of the population, had acquired COVID-19 in the same period [18]. With our intense surveillance and testing strategy, we showed that the infection rate was 23.3% among our staff. The lower infection rate was likely due to a high vaccination rate and more stringent infection-prevention behavior influenced by the participants’ medical background or training. The overrepresentation of females in our cohort and groups in close contact likely resulted in a seemingly increased risk of COVID-19 in female HCWs. The absence of severe cases was likely a result of high vaccination coverage and, more importantly, a relatively young mean age of 37.7 years. The proportion of asymptomatic infection in our cohort was lower than that in studies described previously from South Africa (23%) and China (46.7%) but was similar to a cohort of healthcare personnel from New York (11%) during the Omicron epidemic [19,20,21]. The actual proportion of asymptomatic infection in our cohort could be overestimated, as the clinical data could have been submitted during the pre-symptomatic stage of infection. Although being symptomatic and having an exposure history were the most common reasons for undergoing testing, regular mandatory RAT played an important role in promoting testing, as up to 26.66% of the infected staff were identified as a result of such policy. This could have identified early infection and prevented onward transmission among staff and patients.

Two types of vaccines have been available in Hong Kong since late February 2021: inactivated COVID-19 Vaccine (CoronaVac; Sinovac) and mRNA Vaccine (BNT162b2; Comirnaty). Although BNT162b2 was found to elicit a more robust humoral response and a higher vaccine effectiveness (VE) against symptomatic infection, both vaccines were shown to be effective in preventing hospitalization and death in the pre-Omicron era [22,23,24]. As a result of the large number of amino acid substitutions in the receptor-binding domain of the spike protein, Omicron VOC is capable of evading immunity from previous vaccination or infection [25]. Reduced VE associated with two-dose vaccination and immune waning over time were evident. In South Africa, where Omicron was first identified, the VE of two doses of BNT162b2 was found to decline from 93% during the comparator period to 70% shortly after Omicron had become the dominant strain [5]. A similar decline in VE was observed in different countries when “previously circulating VOC” were taken over by Omicron [3,26]. Real-world data for CoronaVac’s VE against Omicron are scarce. In a study conducted between 6 December 2021 and 26 February 2022 during the Omicron outbreak in Chile, the estimated VE was modest at 38.2% (95% confidence interval (CI), 36.5–39.9) against symptomatic COVID-19 in children 3–5 years of age, although protection against hospitalization and ICU admission remained around 60% [27]. A study from Hong Kong found that two doses of BNT162b2 or CoronaVac vaccines provided an inadequate 50%-plaque-reduction neutralization test (PRNT50) antibody immunity against the Omicron variant. Furthermore, only 1 out of the 30 individuals in the COVID-19 convalescent cohort at 4.8–6.5 months post-symptom onset met the protective antibody threshold for the Omicron variant [28].

To combat the problem of waning immunity and immune escape associated with the Omicron variant, a booster dose is now widely administered in many countries to restore protection against COVID-19. In Hong Kong, based on the latest available evidence and expert opinion, a third dose of CoronaVac can be given to individuals 3 years of age or older, while a third dose of BNT162b2 can be given to individuals 5 years of age or older [29]. The additional protection from three-dose BNT162b2 vaccination is well established, with consistent data from multiple large-scale studies. In the United Kingdom, vaccine effectiveness against Omicron after two BNT162b2 doses declined to 8.8% (95% CI, 7.0 to 10.5) at 25 or more weeks, and a booster dose increased VE to 67.2% (95% CI, 66.5 to 67.8) at 2 to 4 weeks [3]. In Qatar, BNT162b2 effectiveness was highest at 46.6% (95% CI: 33.4–57.2%) against symptomatic BA.1 and at 51.7% (95% CI: 43.2–58.9%) against symptomatic BA.2 infections in the first three months after the second dose, but declined to ~10% or below thereafter. Effectiveness rebounded to 59.9% (95% CI: 51.2–67.0%) and 43.7% (95% CI: 36.5–50.0%), respectively, in the first month after a booster dose [5]. In our study, the lowest COVID-19 incidence in the three-dose BNT162b2 group is consistent with these overseas data.

For individuals who received two doses of CoronaVac, using live virus neutralization assay, heterologous boosting with BNT162b2 was found to induce a better neutralizing antibody titer against Wild-type, Beta, Delta, and Omicron variants than homologous boosting [8,28]. Using a surrogate neutralizing antibody immunoassay, our group has previously demonstrated, in individuals who had negative neutralizing antibody after two doses of CoronaVac (primary non-responder or waned antibody), that BNT162b2 booster induced a significantly higher percentage of positive neutralizing antibody against Delta and Omicron variant than the CoronaVac booster. Using an interferon-gamma release assay, the BNT126b2 booster was also found to induce a better T-cell response [30]. Our current study has provided real-world data on the enhanced protection against Omicron with heterologous boosting after two doses of CoronaVac. We showed that three-dose vaccination significantly reduced the chance of COVID-19, and according to regression analysis, the effect mainly came from the BNT162b2 booster. Large-scale case–control or prospective study is needed to confirm the benefit of mRNA vaccine over inactivated vaccine as a booster.

For infected staff to return to work, we took a more stringent approach by using RT-PCR criteria since the Ct value strongly correlates with the presence of a live virus in individuals with SARS-CoV-2 infection [31]. A study has shown that E gene Ct value of >30 was associated with reduced infectivity and secondary transmission rate [32]. Although negative RATs can be used as a surrogate for reduced infectivity and have been used to end isolation for the general public, the performance of our RAT kit (INDICAID COVID-19 Rapid Antigen Test) has not been thoroughly evaluated with respect to this purpose; moreover, RAT sensitivity could be affected by the sampling technique [33]. We believe a more stringent approach should be taken for recovering healthcare work to prevent nosocomial transmission. In a viral shedding kinetics study of 45 patients infected with the Delta variant, the viable virus in the cell culture was detected for a notably shorter duration in those who were fully vaccinated [34]. A viral dynamic study from the United States and Singapore performed in the pre-Omicron era also showed a shorter viral clearance time in vaccinated individuals [35,36]. However, we were not able to demonstrate any difference in the time required to return to work with different vaccination regimes, nor was it was related to age or gender. We postulate that this could be due to a less effective clearance of the Omicron variant by mismatched antibody induced from vaccines using the wild-type target.

Our study has several limitations. First, our cohort is retrospective in nature with a small sample size and a relatively young age, so the result may not be generalizable to those <18 years of age or the elderly population. Second, because they have a background of being medically trained, our sample may have been more meticulous in terms of infection prevention practices and risk avoidance compared to the general public during social activity or within a household, especially when there is a confirmed/suspected case. Third, despite a well-defined definition for significant exposure, we were not able to further quantify the intensity of exposure, especially in the context of household contact; e.g., continued sharing of a toilet in the same apartment was unavoidable for many while some could temporarily relocate away from the index case. Fourth, the medical history of the participants was not available, although the number of immunocompromised individuals would be extremely small and may not have impacted the final result. Finally, virus-sequencing data were not available and we cannot rule out the possibility of non-Omicron variants in our cohort. Additionally, since the cohort was predominantly in the context of a BA.2.2 outbreak, our findings may not be generalizable to BA.4 and BA.5 variants, which have greater infectivity. In conclusion, we have demonstrated a reduction in breakthrough SARS-CoV-2 infections in healthcare workers with homologous or heterologous BNT162b2 boosting in a territory-wide Omicron BA.2.2 outbreak.

## Figures and Tables

**Figure 1 vaccines-10-01322-f001:**
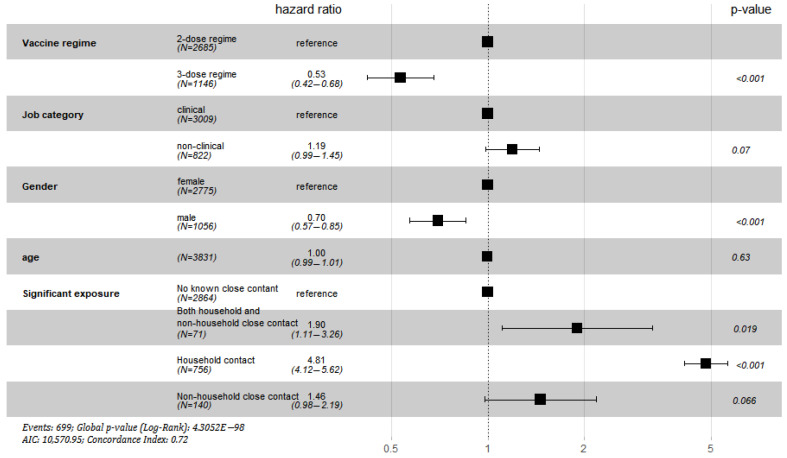
Hazard ratio for COVID-19 and associated 95% confidence interval.

**Figure 2 vaccines-10-01322-f002:**
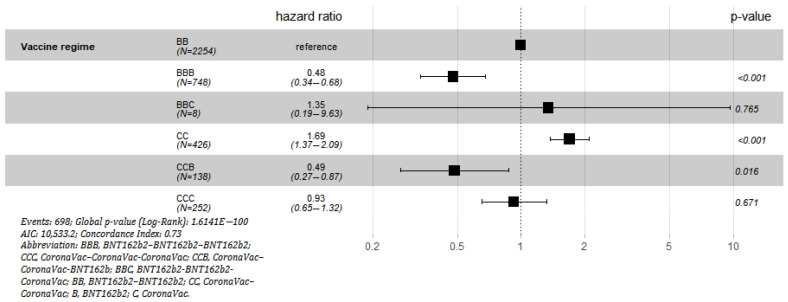
Hazard ratio for COVID-19 and associated 95% confidence interval for different vaccine regime using 2-dose BNT162b2 as reference. (Abbreviation: BBB, BNT162b2–BNT162b2–BNT162b2; CCC, CoronaVac–CoronaVac-CoronaVac; CCB, CoronaVac–CoronaVac-BNT162b; BBC, BNT162b2-BNT162b2-CoronaVac; BB, BNT162b2–BNT162b2; CC, CoronaVac–CoronaVac; B, BNT162b2; C, CoronaVac.).

**Figure 3 vaccines-10-01322-f003:**
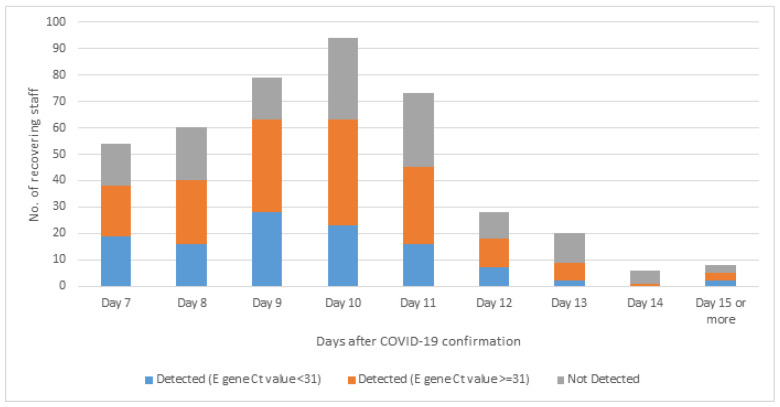
First RT-PCR result after 2 consecutive negative rapid antigen tests in recovering staff who were previously fully vaccinated (N = 422).

**Table 1 vaccines-10-01322-t001:** Vaccination status of 3167 hospital staff before and at the end of the study period.

No. of Doses Received	Total No. of Staff (%) N = 3165 ^#^	No. of Staff (%)				
		Non-Mixed Vaccine Platform		Mixed Vaccine Platform		
**Vaccination status as of 1 February 2022**						
		BNT162b2	CoronaVac	BNT162b2-CoronaVac	BNT162b2-BNT162b2-CoronaVac	CoronaVac-CoronaVac- BNT162b2
1	126 (3.98%)	83 (2.62%)	43 (1.36%)			
2	2439 (77.00%)	2076 (65.59%)	359 (11.34%)	2 (0.06%)		
3	514 (16.24%)	230 (7.27%)	172 (5.43%)		5 (0.16%)	107 (3.38%)
**Vaccination status as of 31 March 2022**						
1	35 (1.11%)	20 (0.63%)	15 (0.47%)			
2	1669 (52.67%)	1419 (44.83%)	245 (7.74%)	3 (0.09%)		
3	1434 (45.31%)	983 (31.06%)	291 (9.19%)		11 (0.35%)	149 (4.71%)

^#^ Including 88 (2.78%) and 29 (0.92%) staff members with zero doses of vaccine as of 1 February 2022 and 31 March 2022, respectively. Two staff members who received mRNA–1273-mRNA-1273 and Sinopharm–CoronaVac were excluded from analysis.

**Table 2 vaccines-10-01322-t002:** Demographics and symptoms of staff with COVID-19 during the study period.

			No. of Staff (%) N = 737
Female			596 (80.9%)
Age (years)	Mean ± SD	37.7 ± 10.5	
	Median	36	
Staff category	Clinical		576 (78.15%)
		Doctor	6 (0.81%)
		Nurse	270 (36.64%)
		Supporting Staff	250 (33.92%)
		Allied Health	50 (6.78%)
	Non-clinical		161 (21.85%)
		Supporting Staff	87 (11.80%)
		Engineer/Technician	21 (2.85%)
		Food and beverage	53 (7.19%)
Positive RAT at the time of COVID-19 confirmation			517 (70.15%)
Positive RT-PCR at the time of COVID-19 confirmation			439 (59.57%)
Having at least 1 COVID-19 related symptom ^#^			649 (88.06%)
Reason for undergoing the index COVID-19 testing *	New onset of COVID-19-related symptom(s)		349 (53.78%)
	Contact with a confirmed case		215 (33.13%)
	Contact with a person with sign(s)/symptom(s) of COVID-19		66 (10.17%)
	Government gazettes compulsory testing notice		16 (2.47%)
	Hospital regular rapid antigen test		173(26.66%)
Symptom(s) reported *			
	Sore throat/throat discomfort		368 (81.06%)
	Cough		275 (60.57%)
	Running nose		212 (46.69%)
	Fatigue		195 (42.95%)
	Headaches		190 (41.85%)
	Fever		188 (41.41%)
	Body aches		158 (34.80%)
	Chills		129 (28.41%)
	Dizziness		57 (12.56%)
	Diarrhea		41 (9.03%)
	Shortness of breath		30 (6.61%)
	Vomiting		18 (3.96%)
	Loss of taste		12 (2.64%)
	Hoarse of voice		6 (1.32%)
	Sputum		5 (1.10%)
	Stuffy nose		5 (1.10%)
	Loss of smell		2 (0.44%)
	Earache		1 (0.22%)
	Bone pain		1 (0.22%)
	Nausea		1 (0.22%)

^#^ At the time of online questionnaire submission. * More than 1 response was allowed. (Abbreviation: RAT, rapid antigen test; RT-PCT reverse transcription-polymerase chain reaction.).

**Table 3 vaccines-10-01322-t003:** Demographics, history of significant exposure and rate of COVID-19 stratified by vaccination regime.

		Vaccination Regime ^#^						*p*-Value ^		
		3-Dose Regime				2-Dose Regime				
		BBB	CCC	CCB	BBC	BB	CC	Comparing 3-dose to 2-dose regime as a whole	Comparing within 3-dose regime	Comparing within 2-dose regime
**COVID-19 positive rate**		4.83%	15.23%	8.70%	12.50%	30.15%	43.01%			
**No. of COVID-19/total vaccinated**		37/766	39/256	12/138	1/8	490/1625	120/279	<0.0001	<0.0001	<0.0001
**No. of female (%)**		514 (67.10%)	159 (62.10%)	91 (65.94%)	5 (62.5%)	1268 (78.03%)	208 (74.55%)	<0.0001	0.5144	0.2141
**Mean age (years)**		42.35	49.88	50.07	46.38	34.89	44.95	<0.0001	<0.0001	<0.0001
**Staff category** **(%)**	Clinical (vs. non-clinical)	588 (76.76%)	194 (75.78%)	107 (77.54%)	7 (87.5%)	1323 (81.42%)	209 (74.91%)	0.0137	0.9352	0.0141
**No. of staff reported significant exposure (%)**	Household contact only	91 (11.88%)	44 (17.19%)	24 (17.39%)	0	427 (26.28%)	80 (28.67%)	<0.0001	0.3979	0.3904
	Non-household close contact only	20 (2.61%)	6 (2.34%)	5 (3.62%)	0	81 (4.98%)	9 (3.23%)			
	Both household & non-household close contact	11 (1.44%)	4 (1.56%)	2 (1.44%)	0	34 (2.09%)	3 (1.08%)			

^#^ 5 cases of BNT162b2-CoronaVac, mRNA-1273-mRNA-1273, Sinopharm-CoronaVac excluded; 90 cases of incomplete vaccination (0 or 1 dose) excluded. ^ Using t-test/Fisher’s exact test. (Abbreviation: BBB, BNT162b2-BNT162b2-BNT162b2; CCC, CoronaVac-CoronaVac-CoronaVac; CCB, CoronaVac-CoronaVac-BNT162b; BBC, BNT162b2-BNT162b2-CoronaVac; BB, BNT162b2–BNT162b2; CC, CoronaVac–CoronaVac.).

**Table 4 vaccines-10-01322-t004:** Time-dependent Cox regression analysis on the risk of acquiring COVID-19.

	Estimate	Hazard Ratio	*p*-Value	95% CI of Hazard Ratio
3-dose vaccination (2-dose vaccination as reference)	−0.6276	0.5339	<0.0001	(0.420, 0.679)
Non-clinical staff (clinical staff as reference)	0.1778	1.1945	0.0700	(0.986, 1.448)
Female staff (male staff as reference)	0.3596	1.4328	0.0005	(1.172, 1.752)
Age	−0.0019	0.9981	0.6296	(0.991, 1.006)
Close contact history (no known close contact as reference)				
- Household close contact only	1.5712	4.8126	<0.0001	(4.121, 5.621)
- Non-household close contact only	0.3789	1.4607	0.0656	(0.976, 2.186)
- Both household and non-household close contact	0.6426	1.9013	0.0193	(1.110, 3.258)

**Table 5 vaccines-10-01322-t005:** Time-dependent Cox regression analysis on effect of 3-dose vs. 2-dose regime on risk of acquiring COVID-19 in household and non-household close contact setting.

	Estimate * (3-Dose vs. 2-Dose)	Hazard Ratio	*p*-Value	95% CI of Hazard Ratio
Household contact only	−0.3814	0.6829	0.0248	(0.490, 0.953)
Non-household close contact only	−2.2282	0.1077	0.0355	(0.014, 0.859)
Both household and non-household close contact **	−0.3925	0.6754	0.652	(0.123, 3.717)
No known close contact	−0.8686	0.4196	<0.0001	(0.293, 0.601)

* Other variables included job category, gender and age. ** All infected are female.

**Table 6 vaccines-10-01322-t006:** Time-dependent Cox regression analysis on risk of acquiring COVID-19 with different vaccination regime *.

	Estimate	Hazard Ratio	*p*-Value	95% CI of Hazard Ratio
Vaccination regime (BB as reference)				
- CC	0.5267	1.6933	<0.0001	(1.370, 2.093)
- BBB	−0.7385	0.4778	<0.0001	(0.336, 0.679)
- BBC	0.2995	1.3491	0.7652	(0.189, 9.627)
- CCB	−0.7211	0.4862	0.0157	(0.271, 0.873)
- CCC	−0.0760	0.9269	0.6715	(0.653, 1.317)

* Other variables including job category, gender, age, and exposure history are not shown. (Abbreviation: BBB, BNT162b2-BNT162b2–BNT162b2; CCC, CoronaVac–CoronaVac–CoronaVac; CCB, CoronaVac–CoronaVac-BNT162b; BBC, BNT162b2-BNT162b2–CoronaVac; BB, BNT162b2– BNT162b2; CC, CoronaVac–CoronaVac; B, BNT162b2; C, CoronaVac.).

**Table 7 vaccines-10-01322-t007:** Association between vaccine regime, gender, age, and time taken to return to work after COVID-19 ^#^.

		N	Mean No. of Days Taken to Fulfil RT-PCR Criteria for Return-to-Work ^#^	*p*-Value *
Vaccine regime	3-dose	423	9.85	0.15
	2-dose		10.20	
	2 or 3 doses BNT162b2	423	10.13	0.8
	2 or 3 doses of CoronaVac		10.08	
	BNT162b2 as 3rd dose	60	9.95	0.865
	CoronaVac as 3rd dose		10.04	
	Last dose within 180 days of COVID-19	423	10.19	0.206
	Last dose > 180 days before COVID-19		9.92	
Gender	Male	423	9.8	0.088
	Female		10.19	
Age	50 years or above	423	10.2	0.676
	Below 50 years		10.1	

* Using t-test. ^#^ COVID-19-recovered staff with negative RT-PCR or a positive test with E gene Ct value 31 or above can return to work.

## Data Availability

The data used to support the findings of this study are included within the article.

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
