# Peer review of "Impact of COVID-19 Vaccination on Healthcare Worker Infection Rate and Outcome during SARS-CoV-2 Omicron Variant Outbreak in Hong Kong"

_vaccines, 2022, doi:10.3390/vaccines10081322_

Round 1
Reviewer 1 Report
The manuscript by Zee et al. presents an analysis of the efficacy of currently available Covid-19 vaccines against the newly circulating Omicron variant of the virus. The study was performed in a relatively large cohort of health-care workers and compares the effect of different combinations of mRNA and inactivated vaccines on the risk of infection. The results are interesting and actual.
The manuscript is well written overall, but especially the Results section deserves better presentation for the reader.
First it should adhere to the Instruction for authors in the following: "All Figures, Schemes and Tables should be inserted into the main text close to their first citation and must be numbered following their number of appearance (Figure 1, Scheme I, Figure 2, Scheme II, Table 1, etc.)." As it is, the flow of the data description in the text and their presentation in the Figures nad Tables is confusing. The section should thus be re-organized and text description adjusted to describe the important findings presented in the adjacent Tables and Figures.
The overwhelming majority of the publication is based on data from individuals vaccinated with either Comirnaty or Sinovac vaccines, or combination of these two. On line 52 the authors state:"Two types of vaccines are available in Hong Kong since early 2021: inactivated 53 COVID-19 Vaccine (CoronaVac; Sinovac), mRNA Vaccine (BNT162b2; Comirnaty).". However, in Table 1, individuals vaccinated with mRNA-1273 and Sinopharm are included - one in each group. This confuses the Table and, since the numbers are low, should be removed both from the presentation and analysis.
The substantive "staff" represents both plural and singular. While in some places throughout the paper it is used as such, in other the authors introduce "staffs". It should be corrected. If the "staff" needs to be exactly counted, it can be either done as, e.g. "8 members of staff", or described otherwise (...8 HCW..)
Author Response
1/ Tables and figures are now re-organized and placed next to the relevant text.
2/ The 2 individuals vaccinated with mRNA-1273 and Sinopharm are now excluded from Table 1. Since they were not included in the analysis of vaccine effect, the subsequent statistical results were not affected
3/ "staffs" were replaced with "staff members"
(All changes are highlighted)

Reviewer 2 Report
This paper reports an interesting study on real life effects of a 3rd vaccination in the protection against infection by the SARS-CoV-2 omicron variant. The study is thorough and based on a large number participants. I have only a few minor points.
1. The six million deaths mentioned in the introduction are the reported deaths. The real numbers are probably much higher. See e.g. https://www.nature.com/articles/d41586-022-00104-8
2. I assume that omicron BA.1/2 was predominant during the study period. Probably good to add this, since most countries are now in or past the BA.4/5 wave.
3. It is clear, from ref. 2 in the paper, that the extra protection by a 3rd vaccination wanes. Should we have a 4th vaccination, as in some countries or do we need more broadly protecting vaccines? Something to mention in the discussion, also in relation to the diminished efficacy of vaccines against BA.4/5.
Author Response
1/ The point of underestimating death rate is now added to the introduction and reference added
2/ We supplemented that the Hong Kong outbreak was caused by BA.2.2 and reference added
3/ The limitation of our finding being not generalizable to BA.4 and BA.5 variants is added to the last paragraph
(All changes are highlighted)
